# Review on PLGA Polymer Based Nanoparticles with Antimicrobial Properties and Their Application in Various Medical Conditions or Infections

**DOI:** 10.3390/polym15173597

**Published:** 2023-08-30

**Authors:** Ashok K. Shakya, Mazen Al-Sulaibi, Rajashri R. Naik, Hamdi Nsairat, Sara Suboh, Abdelrahman Abulaila

**Affiliations:** 1Faculty of Pharmacy, Al-Ahliyya Amman University, Amman 19328, Jordan; 2Pharmacological and Diagnostic Research Center, Faculty of Pharmacy and Allied Medical Sciences, Al-Ahliyya Amman University, Amman 19328, Jordan; 3Faculty of Allied Medical Sciences, Al-Ahliyya Amman University, Amman 19328, Jordan

**Keywords:** PLGA, antibiotics, natural product, magnesium, antibacterial, biomedical application

## Abstract

The rise in the resistance to antibiotics is due to their inappropriate use and the use of a broad spectrum of antibiotics. This has also contributed to the development of multidrug-resistant microorganisms, and due to the unavailability of suitable new drugs for treatments, it is difficult to control. Hence, there is a need for the development of new novel, target-specific antimicrobials. Nanotechnology, involving the synthesis of nanoparticles, may be one of the best options, as it can be manipulated by using physicochemical properties to develop intelligent NPs with desired properties. NPs, because of their unique properties, can deliver drugs to specific targets and release them in a sustained fashion. The chance of developing resistance is very low. Polymeric nanoparticles are solid colloids synthesized using either natural or synthetic polymers. These polymers are used as carriers of drugs to deliver them to the targets. NPs, synthesized using poly-lactic acid (PLA) or the copolymer of lactic and glycolic acid (PLGA), are used in the delivery of controlled drug release, as they are biodegradable, biocompatible and have been approved by the USFDA. In this article, we will be reviewing the synthesis of PLGA-based nanoparticles encapsulated or loaded with antibiotics, natural products, or metal ions and their antibacterial potential in various medical applications.

## 1. Introduction

It is estimated that around 700,000 people die every year due to resistant microorganisms all over the world [1,2]. Resistance to antimicrobials is caused when microorganisms can grow in the presence of antimicrobial agents [3,4]. A list of antibiotic-resistant bacteria, both Gram-positive and Gram-negative bacteria, was published in 2017 [5]. This rise in the resistance to antibiotics is mainly attributed to the inappropriate use of antibiotics and the use of a broad spectrum of antibiotics, which has led to the emergence of multidrug-resistant microorganisms. Moreover, this has led to difficulties in treating the disease caused by multiple drug-resistant organisms; due to the unavailability of suitable new drugs, it is difficult to treat the epidemic pathogens. Therefore, there is a need for the development of new, novel antimicrobials with improved targeted activity toward these microorganisms. One of the best options is to use nanotechnology. By using nanotechnology in the medical field, NPs can be created or developed to either mimic or alter biological processes [6]. Nanoparticles vary in size, with an ideal size of 10 to 1000 nm; the size of 100 to 500 nm is used in medical applications [6,7]. Well-defined specific NPs with increased bioavailability can be developed by manipulating the physicochemical properties and the materials used to synthesize the NP [6,8,9,10,11]. It may be noted that NPs offer many advantages, such as increased stability of the hydrophobic drugs, making them suitable for oral administration; they also offer increased bioavailability, improved distribution and pharmacokinetics, as a result of which they reduce side effects, increased efficiencies, they favor the accumulation at the target and, most importantly, reduce the toxic effect by incorporating the biocompatible polymers [12]. As mentioned earlier in the paragraph, inappropriate use of antibiotics can result in the bacteria developing resistance; to kill the pathogen, a higher dose of antibiotics may be required. This may cause adverse effects, such as alterations in the growth and development of harmless or useful bacteria (in the human intestine), and it may produce toxic effects due to its accumulation at non-target sites [13,14]. One good example of such a toxic effect is the one produced by aminoglycosides and tobramycin administration; prolonged administration of high doses caused nephrotoxicity, which resulted in reduced rate of glomerular filtration that subsequently lead to the alteration of the excretion of electrolytes [15,16].

### 1.1. Challenges in the Delivery of Antimicrobials

To overcome the challenges of delivering antimicrobials to treat bacterial infections, antibiotics are often used as first-line drugs. There are several challenges in the delivery of these antibiotics or in antibiotic therapy. Some of these challenges include resistance to antibiotics, low bioavailability, adverse side effects and further hindrance in the treatment is enhanced due to the presence of physiological barriers.

Resistance to antibiotics may be caused by modifications to the antibiotics that are facilitated by the help of enzymes (causing steric hindrance to its target) or hydrolysis of the drug itself (β-lactamase that breaks the amide bond of the β-lactam ring), rendering it unable to inhibit the target [17,18]. It can also develop resistance by decreasing the penetration of the drug and increasing the efflux of the drug. Most of the targets are intracellular; hence, entry of the drug through the membrane is facilitated by the water-filled diffusion channels porins. This process can be affected by the downregulation, modification or loss of the function of the porin genes that affect the entry of the drug [19]. Similarly, the efflux of these drugs can be increased by the upregulation of active transport through efflux pumps [20]. Further, the accumulation of a mutation can lead to complete replacement of the targets, such as the production of targets that carry out all the biochemical reactions of the target but are not inhibited by the antibiotics; this can be seen in the mutation of the target enzymes that cause bacterial resistance to quinolones [21]. Once established, it may be spread widely among the susceptible bacteria through horizontal gene transfers such as transformation [22], transduction [23] and conjugation [24] and facilitate its uptake among the susceptible bacterial population.

To overcome the resistance, new antibacterial agents with different mechanisms have to be considered; it should also be noted that while considering traditional or novel therapies, the mechanism of antibiotic resistance must be taken into consideration.

One of the other challenges, as mentioned above, is its low bioavailability; antibiotics are administered either orally or through an intravenous route, which results in systemic distribution with only small amounts reaching the infected area. For example, oral administration of fluoroquinolones is extracted through the biliary system, and one-third is found in stools after oral administration [25]. In order to achieve a therapeutic effect, higher doses for longer periods of time must be used; this may cause adverse effects and resistance [26]. On the contrary, if low doses are used as higher doses are not feasible, the pathogenic bacteria may reach a subtherapeutic concentration, whereas the more susceptible bacteria may be prone to adaptive mutation and genetic changes, increasing the risk of resistance to the drug [27]. The adverse effects due to high doses can limit its clinical uses; example includes—high doses of nitrofuran cause pulmonary toxicity, linezolid—hematologic toxicity, metronidazole—neurotoxicity, fluoroquinolones—may increase the risk of aortic aneurysm [28] and gentamicin—kidney injury [29].

In addition to the direct effect on the patients, higher doses of antibiotics may disturb the gut microbiota, which consists of commensal bacteria and inhibit the invasion of pathogenic bacteria and its colonization; it may also cause an imbalance in the gut microbiota and increase its susceptibility to infectious diseases [30,31]. Damage caused by antibiotics to the gut microbiota is long-lasting; this may be observed in short-term exposure to clindamycin, that leads to long-term damage to the gut microbiota.

Challenges caused by the barriers may be categorized as depending on the route of administration. To overcome all these challenges, we can, fortunately, use nanotechnology. Nanotechnology ensures targeted delivery with bioavailability and fewer side effects. Nanotechnology also ensures the stability of the loaded drug.

Hence, a drug delivery system that can overcome the above-mentioned drawbacks, which would be beneficial in improving the effect of the antibiotic, its delivery to the infected site, reducing its accumulation at off-target sites, and overcoming resistance, is required. Nanoparticles are a good choice because of their physicochemical properties and the flexibility they provide in manipulating their physicochemical properties toward synthesizing intelligent NPs. Nanoparticles, because of their unique physical properties, are capable of releasing the drug in a controlled and sustained fashion. Burgess et al. and Shaaban et al. were able to deliver the drug to the infected site, leading to a lower chance of bacteria developing resistance [32,33,34,35,36], which in turn gave less chance for the drug to accumulate at nontarget sites [13,14]. Zhao and Stenzel (2018) and Behzadi et al. (2017) found that nanoparticles also shielded the drug from degradation and allowed the uptake of the drug by bacteria through different routes, compared to the free antibiotics [37,38]. There are several pieces of literature available that highlight the beneficial effect of encapsulating antibiotics using polymer-based nanoparticles to deliver drugs. Polymeric nanoparticles are solid colloids or polymeric compilation synthesized using either natural or synthetic polymers; these polymers are often used as the carriers of a drug, delivering it to the target. They are classified as nanosphere, where the drug is dispersed throughout the matrix [39], and nano-capsules, where the drug is in the core either in the oil or aqueous solution [40] (Figure 1).

NPs can be synthesized using either natural or synthetic polymers. NPs synthesized using natural polymers are usually used in medical applications, although they have some limitations, such as their purity variations; they also require regular crosslinking to enhance their stability [41,42]. NPs synthesized using poly-lactic acid (PLA) or the copolymer of lactic and glycolic acid (PLGA) are used for the delivery of the controlled drug release, as they are biodegradable, biocompatible and have been approved by the USFDA and the European Medicine Agency for the parenteral route of administration [40,41,42]. PLA is used in lesser content because of its low degrading rate compared to PLGA.

### 1.2. Poly(lactic-co-glycolic Acid) (PLGA)

PLGA is a biodegradable polymer and is the most commonly used polymer for drug delivery. It is hydrolyzed to its monomer’s lactic acid and glycolic acid. These monomers are endogenous and utilized by our body through the citric acid cycle [43]. The molecular weight of the PLGA affects the mechanical strength. The molecular weight of the PLGA directly affects the properties of NPs, such as size, the capacity of entrapment, its release ability and biocompatibility [44]. Due to its characteristics, poly(lactic-co-glycolic acid) (PLGA) is widely used as a matrix for NPs. PLGA is obtained by various combinations of lactic acid and glycolic acid during polymerization, as a result of which different molecular weights and physicochemical properties are achieved. The use of PLGA for the delivery of drugs and biomedical applications is associated with minimal systemic toxicity [40]. PLGA is commercially available in different compositions of copolymer and with different molecular weights. Depending on the ratio of the copolymer, the degradation time varies from months to years. The different forms of PLGA are recognized depending on or based on the composition of the copolymer [45]. Internalization of PLGA NPs occurs partly by pinocytosis (cellular drinking) and through clathrin-mediated endocytosis; once inside the cell, they evade the lysosome and reach the cytoplasm within 10 min after incubation. This enables its interaction with the vesicular membrane, causing destabilization of the membranes and its release into the cytosol [46]. One of the processes that accounts for the biological barrier of the PLGA-based NPs is the recognition of the hydrophobic molecules as foreign molecules by the body. The reticuloendothelial system (RES) distributes them from the bloodstream to the liver and spleen. This process is one of the limitations of the PLGA-based NPs controlled drug delivery system [40]. The opsonin proteins in the blood serum recognize these particles and bind to them; these opsonized particles bind to the membranes of the macrophages and are internalized by phagocytosis [40,47]. These limitations can be solved by surface modification of the NPs so that the RES cannot be recognized. Surface modifications may involve coating the surface of the NPs with a hydrophilic layer that hides the hydrophobicity of the NPs. One of the most commonly used chemical moiety in surface modifications is the hydrophilic and nonionic polymer polyethylene glycol (PEG), which increases its half-life in blood circulation; PEG is biocompatible [47] with other polymers that are used in surface modifications, including poloxamer, poloxamines, or chitosan [39]. These are known to block the electrostatic and hydrophobic interactions that allow opsonin proteins to bind. PEGylation of PLGA NPs or surface modification using chitosan can shift the negatively charged PLGA to neutral or positively charged NPs so that the positively charged NPs can escape from the lysosome after internalization and localize to the perinuclear region [48,49]. Neutral and positively charged particles co-localize in the lysosomal region [50].

PLGA is used in the encapsulation of several hydrophilic and hydrophobic drugs such as gentamicin [42], sparfloxacin [51] and doxorubicin [52]. It may be noted that the literature on the review of the synthesis of PLGA polymer-based nanoparticles is very limited. Few studies have been conducted on the formulation of nanoparticles employing (PLGA) polymers, polymer nanoparticles with antimicrobial agents or as carriers for antimicrobial agents, or polymer-based nanoparticles with antimicrobial drugs. Hence, in the present review, an attempt has been made to review the synthesis of PLGA-based nanoparticles with antibacterial properties and poly ethylenediamine nanoparticles (PNP) used in various medical applications [53,54].

## 2. Synthesis of Antibiotics Encapsulated in PLGA Nanoparticles

### 2.1. Clarithromycin (CLR)

The chemical formula of CLR is C_38_H_69_NO_13_ and its structure is depicted in Figure 2. It is used in the infection of various organs such as the upper and lower respiratory tract, skin, ear, and soft tissue. It is a semi-synthetic macrolide antibiotic used to prevent various bacterial infections. It is administered twice daily, with a half-life of 3–4 h. CLR has low systemic oral bioavailability and is stable in acid. The chemical or therapeutic properties, such as low systemic oral bioavailability and short half-life of the drug, limit the therapeutical potentials of CLR in intracellular infection. In such a case, a high dose and prolonged administration may be required to accomplish its therapeutic potential, which may have toxic side effects such as hepatotoxicity [55,56].

There are several studies that used PLGA nanoparticles to load CLR, and these studies have employed different ratios of PLGA to synthesize NPs [57,58,59]. In one of the studies, NPs were synthesized using the nanoprecipitation technique, using polymer ratios, and only Resomer^®^ RG 502 (lactic acid:glycolic acid ratio 50:50, average Mw: 12,000) [58]. In another study, CLR-loaded PLGA-based NPs were synthesized using only one type of polymer (PLGA) with a ratio of (lactic acid: glycolic acid ratio 75:25, Mw 15,000–30,000); they used a modified O/W single emulsion solvent evaporation technique [59]. Jain and his colleagues synthesized PLGA-based NPs loaded with CLR using a solvent evaporation technique. In their study, they utilized only one type of polymer (lactic acid:glycolic acid ratio 50:50) with a different drug [57]. None of the studies mentioned above have studied or emphasized the effect of the molecular weight of the PLGA polymer on CLR-loaded NPs [52]. Oztürk and his coworkers synthesized CLR-loaded PLGA-based NPs using the nanoprecipitation technique. To increase oral bioavailability and enhance its therapeutic potential as an antibacterial agent, the authors used three different molecular weights of PLGA to synthesize the NPs and three different chitosan-coated NPs [60] (Table 1).

To synthesize PLGA-based nanoparticles, 90 mg of PLGA was dissolved in 3 mL of acetone along with Span^®^ 60 (30 mg). Then, 3 mL of the dissolved solution was added drop by drop at a rate of 5 mL H^−1^ into 10 mL of Pluronic^®^ F-68 aqueous solution (0.5%, *w*/*v*) under a magnetic stirrer. Then, acetone was added and mixed by using a magnetic stirrer for 4 h at room temperature. The resulting solution was subjected to centrifugation at 11,000 rpm for 45 min at 4 °C. The synthesized NPs were collected and 5 mL of distilled water was added to clean the NPs before centrifugation; this was repeated twice. For the synthesis of the CLR-loaded PLGA NPs, they started the procedure by dissolving 9 mg of CLR in the organic phase, followed by the same procedure as said above; 3 mL of this solution was added in a dropwise manner at a slow rate of 5 mL H^−1^ to a Pluronic^®^ F-68 aqueous solution (0.5%, *w*/*v*) with magnetic stirring; the acetone was evaporated under room temperature using a magnetic stirrer for 4 h. The resulting solution was subjected to centrifugation (11,000 rpm, 45 min, 4 °C), followed by the collection of the NPs and washing with distilled water; the process of centrifugation was repeated two to three times. For the synthesizing of the CS-coated formulations, they followed the same procedure with a few modifications.

They tested the antibacterial potentials of the synthesized NPs on *Staphylococcus aureus* (ATCC 25923), *Enterococcus faecalis* (ATCC 29212), *Listeria monocytogenes* (ATCC 1911), and *Klebsiella pneumoniae* (ATCC 700603). Most of the synthesized compounds exhibited antibacterial activity against the tested microorganism. Only *Enteroccous faecalis* (ATCC29212) was not susceptible to the synthesized compounds. They concluded that the synthesized nanoparticles (503H, 504H and CS-504H) showed remarkable antibacterial activity against *Staphylococcus aureus*.

### 2.2. Rifampicin (Rif)

Rifampicin is an antibiotic used to prevent various infections caused by the Mycobacterium genus, including the *Mycobacterium avium* complex, such as leprosy and to treat tuberculosis in combination with other drugs. It is a semisynthetic drug synthesized by *Streptomyces mediterranei*. Rifampicin inhibits DNA-dependent RNA polymerase activity, there by inhibiting the synthesis of RNA. It is bactericidal in action against both Gram-positive and Gram-negative bacteria. The structure of the drug is depicted in Figure 2. Maghrebi and her research team improved the potency of this antibiotic against the intracellular small colony variants (SCV) of *Staphylococcus aureus* through an encapsulation strategically engineered cell penetrant delivery method [61]. They synthesized lipid nanoparticles with a PLGA matrix. PLGA—lipid hybrid (PLH) microparticles by a spray drying process. They loaded the antibiotics rifampicin in both polymer and lipid phases to create a nanoparticle–in–microparticle system that has the capacity to disperse into the media with controlled release.

For the synthesis of rifampicin-loaded PLGA nanoparticles (Rif-PLGA), they dissolved the drug in different ratios (1:1, 1:2, 1:5, 1:10 and 1:20 *w*/*w*) with PLGA. PLGA dissolved along with rifampicin (10 mg) in ethyl acetate (2 mL) with continuous stirring (2 h) at room temperature. An organic solvent was added to 20 mL of aqueous solution of evaporated PVA. Rif-PLGA nanoparticles were collected after centrifugation and washed with 2 mL of distilled water. The washed pallet was redispersed in water and frozen for 48 h in a deep freezer (−20 °C) and later freeze-dried. In their study, they used PLGA microparticles as a control. For this, they dissolved 10 mg and 200 mg of PLGA in 2 mL ethyl acetate; the oil phase was added to PVA (1% *w*/*v*, 10 mL) with stirring, it was evaporated, centrifuged, and the pellets were collected and washed two to three times and redispersed in 2 mL water. The pellets were kept in the freezer (for 48 h) for freeze-drying to achieve PLGA microparticles [61]. Maghrebi et al. (2020) concluded in their study that when compared to Rif-PLGA microparticles and the rifampicin solution, a 4- to 7-fold increase in the uptake of rifampicin was observed in the Rif-PLH microparticles. They attributed this to the enhanced phagocytic properties of the PLH microparticles. This enhances the release of the drug into the inside of the cell, resulting in a 4-fold decrease in *S. aureus* SCV compared to the rifampicin solution. The actual dose of rifampicin in PLH microparticles was 2.5 μg/mL, which caused a 2.5 log reduction in colony forming unit. They emphasized that the increased efficiency, along with the intracellular delivery of the drug by PLH microparticles, may reduce the long-term systemic approach of treating with high doses, reducing toxicity and resistance to antibiotics [61].

### 2.3. Ciprofloxacin (CIP)

Ciprofloxacin is a fluoroquinolone antibiotic; it is a broad spectrum antibiotic that is used in many bacterial infections. The mechanism by which it prevents bacterial infection is by the inhibition of the enzyme DNA gyrase that is involved in DNA replication. It is active against both Gram-positive and Gram-negative bacteria [62]. Gheffar et al. synthesized PLGA polymer based nanoparticles loaded with the antibiotic ciprofloxacin by using the nanoprecipitation method [63]. They studied the antibacterial activity of the synthesized particles on both the planktonic and biofilm modes of *S. aureus* cells (ATCC 29,213 strain). Their studies indicated that the antibacterial potentials of the synthesized nanoparticles with an average particle size of 60 nm, even with the presence of serum protein, were able to completely remove the planktonic culture after 24 h. However, the CIP loaded NPs had low antibacterial activity against the planktonic culture when compared to the free antibiotic due to the sustained release of the drug. The antibiofilm activity of the CIP loaded NPs was significantly higher than the CIP (free antibiotic), this is attributed to the penetration of the NPs into the matrix with the subsequent release of the drug into the bacterial region [63] (Figure 3).

### 2.4. Thiosemicarbazone (TSC)

Molecular components with enhanced pharmacological activities are antioxidant, antibacterial, anticancer, antitumor, antimalarial, antifungal, and inhibit enzyme ribonucleotide reductase, either in coordination with metals or without. TSC is synthesized by the condensation reaction of thiosemicarbazides with aldehydes or ketones.

Hossein Barani and his colleagues developed antibacterial fiber mats using PLGA and different amounts of TSC using the electrospinning technique for the synthesis of electrospun fiber mats [64]. They prepared the PLGA solution by dissolving 0.2 g of PLGA with 7.05 mL of TFE (trifluoroethanol) and constantly stirring for 4 h. different amounts of N4-(S)-(1-phenethyl)-2-pyridin-2-yl-methylene)hydrazine-1carbothioamide (HfpyTSCmB) (0, 2.5, 5, and 10 wt.% relative to PLGA polymer), were added along with the polymer in the TFE solution. The fiber mats were spun using the electrospinning technique. These fiber mats were tested for their antibacterial activity using the zone-of-inhibition technique. The PLGA–TSC mats exhibited an inhibition zone against both Gram-positive (*Staphylococcus aureus*) and Gram-negative (*Escherichia coli*) bacteria with 0.5 to 1.5 mm, while only the fiber mats did not exhibit antibacterial activity. They concluded that these mats might be beneficial in preventing infection in wounds and could be used in wound healing.
polymers-15-03597-t001_Table 1Table 1PLGA/PLA/Chitosan nanocomposites and their biological application.SN.Polymers and Auxiliary Material UsedAntibiotics UsedDescription of FormulationBiological ActivityReferencesYear1PLGAHalloysite nanotubes (HNTs)Hydrophilic chitosan nanofibersAmoxicillinHalloysite nanotubes loaded with amoxicillin—incorporated with PLGA and electrospun with chitosan nanofibers.Wound dressings[65]20162Poly(lactic acid) (PLA)--Norfloxacin-tenoxicamPLA microsphere loaded with Norfloxacin-Tenoxicam was synthesized and studied for its antibacterial properties.Antibacterial activity[66]20223PLGAPeptide (BAR)-Avidin-palmitylated PLGA NPs.PLGA NPs with modified surface—were loaded with BAR peptides in order to enhance their inhibitory efficiencies.Antibiofilm[67]20154PLGA--AzithromycinPLGA NPs loaded with Azithromycin were synthesized using nanoprecipitation technique.Improving anti-biofilm effects of azithromycin[68]20195PLGA--AzithromycinAzithromycin was encapsulated with PLGA NPs by nanoprecipitation method.Antibacterial activity[69]20226PLGA--Cefaclor monohydrateUsing nanoprecipitation technique, PLGA NPs were loaded with Cefaclor monohydrate (CEF).Antibacterial activity[70]20217PLGA--CefquinomeUsing spray drying technique, microsphere was encapsulated with cefquinome.Lung inflammation[71]20178PLGAhydroxyapatite-Ceftriaxone/cefuroximeBy using MAPLE technique, thin films of (HAp/PLGA) with ceftriaxone/cefuroxime antibiotics (ATBs) were synthesized.Prevention of bone implant-associated infections[72]20169PLGAPolyethylene glycol (PEG)-CiprofloxacinPEGylated particles containing ciprofloxacin were evaluated against S. aureus in planktonic and biofilm mode.Antibacterial activity[63]202110PLGA--CiprofloxacinNanocarriers were prepared by nanoprecipitation method to deliver ciprofloxacin.Antibacterial activity[73]202311PLGASilk fibroin (SF)-Vancomycin hydrochlorideVancomycin encapsulated with PLGA microsphere was synthesized, and its antibacterial properties were evaluated.Antibacterial activity[74]202212PLGA-
CiprofloxacinPLGA-based microsphere and NPs are evaluated against *S. aureus* and *P. aeruginosa*, formulated by emulsification and evaporation methods.Bacterial Biofilms[75]201613PLGA--CiprofloxacinPLGA fibrous mats loaded with ciprofloxacin—synthesized by electrospun technique—modified with hydrophilic sodium alginate (ALG) microparticles.Wound healing[76]201814PLGA-ChitosanClarithromycinPLGA NPs and Chitosan NPs were synthesized using nanoprecipitation technique for oral release.Antibacterial activity[60]201915PLGAPolyethylen-imine-ClindamycinClindamycin—encapsulated PLGA PEI NPS (Cly/PP NPs)— with different charges are synthesized and evaluated against Methicillin-Resistant *Staphylococcus aureus.*Management of MRSA-infected wounds.[77]201916PLGA--Doxycycline-Antibacterial activity[78]202217PLGAGelatin nanofibers-Doxycycline hyclate (DCH)DEX@MSN incorporated into PLGA/ge/nanofiber—fabricated by electrospinning technique to make loose layer.Osteogenic and Antibacterial activities[79]201918PLGA--Vancomycin, ceftazidime, and ketorolacTo fabricate the membranes, electrospinning and co-axial electrospinning process was used.Antibacterial  [80]202119PLGA--EnoxacinEnoxacin-loaded PLGA was fabricated by coating on magnesium scaffold (Enox-PLGA-Mg).Inhibition of Osteoclastic bone resorption and antibacterial[81]201620PLGA--GentamicinGentamicin-loaded PLGA (Gent-PLGA-Mg)—fabricated by coating on magnesium scaffold for sustained release of the drug.Antibacterial [82]201521PLGA--GentamicinMicroparticles of Gentamicin—PLGA incorporated into calcium-phosphate bone substitute.Prevention of maxillofacial and orthopedic implant infections[83]201622PLGASodium carboxymethyl cellulosePorous hydroxyapatite (HAp) bone scaffold.GentamicinTo increase the binding ability of the PLGA microsphere and hydroxyapatite (HAp) bone scaffold, sodium carboxymethyl cellulose was used as cross-linking agent.Antibacterial activity[84]202123PLGAChitosan-GentamicinGentamicin PLGA NPs- surface modified by chitosan.Therapeutic benefits.[85]202024PLGAChitosan-LevofloxacinUsing single emulsification (O/W) solvent evaporation method. Levofloxacin-encapsulated PLGA NPs coated with CS were synthesized.Antibacterial activity[86]202025PLGA--LevofloxacinPLGA microparticles and scaffold with levofloxacin at significant therapeutic concentration for sustained release were synthesized.Antibacterial activity[87]202226PLGAChitosan-MoxifloxacinFabrication of nanoparticles with the help of ultra-sonication was carried out, and after centrifugation, NPs were collected.Antibacterial activity[88]201727PLGA--MoxifloxacinSilica NPs containing calcium, magnesium, copper and strontium are incorporated into PLGA coating—induced angiogenesis and osteogenesis.Bone tissue regeneration [89]202328PLGA--MoxifloxacinNPs are formulated via nano-emulsion, and transferrin was used for surface modification.Against Sepsis[90]202329PLGA--NorfloxacinNanocoating containing PLGA and Norfloxacin was synthesized.Antibacterial activity[91]201930PLGA-Poly-ε-caprolactonetobramycinFor chronic osteomyelitis—3D printed antibiotics encapsulated with poly- ε-caprolactone/poly(lactic-co-glycolic acid) scaffold were developed.Treatment of chronic osteomyelitis[92]201531Hyaluronic acidPLGA-Polymyxin BFor synthesis of HA@PLGA-PMB with enhanced surface properties, hydrophilic hyaluronic acid (HA) is mixed with oil in water containing PLGA copolymer of PMB.Lungs infections[93]202232PLGA--RifampinRifampin was loaded in PLGA NPs matrix.Antibacterial activity[61]202233PLGA--Tilmicosin and clarithromycinPLGA microparticles and biopolymer beads were studied.Antibacterial activity[94]202334PLGAZnOPCL-PLGA nanofiber.Antibacterial[95](2018, pending)35PLGAepsilon-PL--Nanofiber.Antibacterial[96]202336PLGA--CiprofloxacinNanoparticles.Antibacterial activity[97]2007

### 2.5. Minocycline, Metronidazole, and Ciprofloxacin (MMC)

#### 2.5.1. Minocycline

Minocycline is a semi-synthetic derivative of tetracycline, having the same properties but with fewer side effects. It is a broad spectrum antibiotic with a bacteriostatic effect; it causes cell wall loss and inhibits protein synthesis in both Gram-positive and Gram-negative bacteria. It is now available in several topical forms and is used in the treatment of periodontal therapies and to treat bacterial infections [98]. It is used in respiratory, urinary tract infections, and skin infections. It is also used to treat gonorrhea and syphilis, to name a few.

#### 2.5.2. Metronidazole

Metronidazole is a commonly used antibiotic, belonging to the nitroimidazole class of antibiotics. It penetrates the cell membrane and interferes with DNA; it is known to bind to DNA and disrupt it, resulting in cell death. It is bactericidal in action on anaerobic periopathogenic microorganisms such as *prevotella intermedia*, *porphyromonasgingivalis*, *Treponema denticola* and spirochetes. Metronidazole under in vitro conditions is not effective against *Aggregatibacter actinomycetemcomitans*. When used to treat juvenile periodontitis, a condition thought to be linked to *Actinobacillus actinomycetemcomitans*, metronidazole, long known to be ineffective in vitro against the latter bacterium, it was marginally more effective than tetracycline [99]. This emphasizes the multi-infectious character of periodontal disease where in vitro tests do not necessarily reflect in vivo effects [100,101].

#### 2.5.3. Ciprofloxacin

Ciprofloxacin is a synthetic fluoroquinolone that is bactericidal in action, and it is effective against multiplying and resting Gram-negative bacteria. It is given along with metronidazole in the treatment of mixed infections due to its limited action against Gram-negative pathogens. It is active against bacteroides such as Fusobacteria and *Aggregatibacter actinomycetemcomitans,* and it is used for periodontitis due to its chemical stability and fewer side effects [100]. Torshabi et al. synthesized an MMC-loaded PLGA-based microsphere using a double emulsion technique and studied its antibacterial potential against *Aggregatibacter actinomycetemcomitans* [102].

Nojehdehian et al. synthesized PLGA microspheres using solvent evaporation, the water-in-oil-in-water technique [103]. Moreover, 100 mg of PLGA dissolved in 1 mL chloroform was combined with 2 mL of 2% (*w*/*v*) PVA solution in distilled deionized water (ddH_2_O) that contained antibiotics (triple MMC mix, 0.1 g in 1.5 mL); after homogenization for 2 min, it was added to 30 mL of 0.2% (*w*/*v*) PVA/ddH_2_O. Microspheres were collected after centrifugation, which was followed by washing them with ddH2O. The centrifugation process was repeated, and then, after collection, they were stored in a freeze-dried state for 2 days. They used a drug-free PLGA microsphere as a control. They studied the antibacterial activity of the MMC-loaded microspheres through a zone of inhibition test. The MMC microspheres showed significant antibacterial activity for up to 11 days against *Aggregatibacter actinomycetemcomitans*. Krayer J.W. reported that systemic administration of metronidazole in combination with ciprofloxacin may be effective in the treatment of periodontal disease caused by *Aggregatibacter actinomycetemcomitans* [104]. Chuensombat et al. also reported that the MMC microspheres may be more effective against a wide variety of microorganisms when compared to their use against microorganisms when given alone. They concluded that the combination of these three drugs may be effective in treating periodontal disease [77].

### 2.6. Clindamycin (Cly)

Hasan et al. developed bacteria-targeted clindamycin-loaded polymeric NPs to treat wounds infected by MRSA [77]. Clindamycin is FDA-approved for the treatment of MRSA infection. Clindamycin is a semi-synthetic derivative of lincomycin that inhibits ribosomal translocation or inhibits protein synthesis [105]. The MRSA-infected wound can be treated orally, through tablets/suspension, systemically, through an IV or topically, through a gel or cream [106,107]. These approaches have their own limitations, such as adverse effects, improper distribution of antibiotics to bacteria and the possibility of bacteria developing resistance [108]. Hence, Hasan et al. developed an alternate systemic approach to treat the MRSA-infected wound. They synthesized clindamycin-loaded polymeric NPs to treat the wound infected by MRSA. They synthesized PLGA-PEI polyethyleneimines (PEI) NPs (PP NPs), Cly/PP NPs and Cly/P NPs, using an oil-in-water (*o*/*w*) emulsification solvent evaporation method [109,110]. Moreover, 200 mg of PLGA was mixed with 20 mg of either PEI or clindamycin and dissolved in 10 mL dichloromethane. They used 0.5 g of Nile red for fluorescence labeling. The organic phase was poured into 30 mL of 1% PVA and emulsified at a high speed for 2 min using an ice bath, followed by sonication in an ice bath at 159 W (3 min). The mixture was stirred, and the residual solvent was removed. NPs were collected after centrifugation and washed in distilled water two to three times (Nurhasni Hasan et al.).

The adhesion of Cly/PP NPs to negatively charged particles of the cell wall was higher and more efficient in killing bacteria compared to Cly/P NPs. They observed that when Cly/PP NPs were applied to the wound, they were effective in reducing the bacterial growth and the size of the wound. They concluded that the development of positively charged clindamycin-loaded polymeric NPs that target the negative charge on the cell wall may be beneficial in preventing infection and enhancing the healing process and may be used in several infections related to the skin.

### 2.7. Platensimycin (PTM)

Bacterial infections caused by multidrug-resistant pathogens have resulted in increased human deaths and have put human health in great danger. It has been proved that the use of nano antibiotics or nanocarriers for antibiotics is beneficial in delivering the antibiotics to the target. Liu et al. (2020) hypothesized that the pharmacokinetic properties of PTM can be enhanced with the help of a nano-based drug delivery system [111]. They synthesized PTM-loaded PLGA NPs or PAMAM NPs and reported that PTM-loaded PLGA NPs and Poly(amidoamine) (PAMAM) NPs were effective in vitro and in vivo against MRSA. Poly(amidoamine) (PAMAM) are dendrimers that are hyper branched and used for the delivery of drugs such as sulfamethoxazole [112,113] and prulifloxacin [114].

Platensimycin (PTM) is an inhibitor of bacterial fatty acid synthases FabB/FabF. PTM effectively inhibits the growth of Gram-positive bacteria and against MRSA and VRE (vancomycin resistance enterococci) without any observed toxic effect and no cross-resistance in in vitro studies [115,116]. Moreover, it inhibited the growth of MRSA in mouse peritonitis. However, its use has been limited due to its poor pharmacokinetic properties. Liu et al. synthesized PTM-loaded PLGA or PAMAM NPs to improve the poor pharmacokinetic properties of PTM [111] (Figure 4) by using the previously reported emulsification solvent evaporation method [117,118]. To synthesize PLGA/PTM nanoparticles, they dissolved around 60 mg of PTM and 200 mg of PLGA in a mixed solvent of 110 mL dichloromethane: methanol (=7:1, *v*/*v*). This solution mixture was added to 30 mL of 2% PVA (in 0.01 M PBA (pH 7.4), followed by sheared homogenization, sonication and evaporation of the organic solvent. PLGA/PTM NPs were obtained. For the synthesis of PAMAM/PTM NPs, the same procedure was utilized. Moreover, 10 mg of PAMAM and 20 mg of PTM were dissolved in 5 mL methanol and sonicated. Then, this mixture was added to 30 mL of deionized water, followed by evaporation of the organic solvent, and the NPs were collected. They then studied the in vitro and in vivo antibacterial properties of the synthesized nanoparticles. PTM-loaded NPs were effective in inhibiting the biofilm formation in *S. aureus*, and they killed most of the *S. aureus* bacteria in the infected macrophages compared to PTM alone. The poor pharmacokinetic performance exhibited by PTM increased and showed increased AUC_0–t_ (area under the curve) (∼ four and two times) when PTM was encapsulated with PLGA and PAMAM/PTM. The survival chance of the MRSA-infected mice increased when these infected mice were injected intraperitoneally with PTM-loaded NPs; moreover, in this circumstance, all the mice survived, while the infected mice, when injected with PTM alone, died. They stated from their study results that the PTM-loaded NPs improved the poor pharmacokinetic properties of PTM and suggested that this supports the idea of developing bacterial fatty acid synthase inhibitors as one of the promising antibiotics against drug-resistant bacterial pathogens.

### 2.8. Azithromycin (AZI)

Azithromycin is a macrolide antibiotic. Due to its safety profile and effectiveness against a wide range of bacterial infections, parasites and helminths, it is termed a wonder drug [119]. AZI has antimicrobial and immunomodulatory properties, which led to it being suggested as one of the potential therapeutic agents against the SARS-CoV-2 virus [120,121]. It may be further emphasized that AZI was biannually administrated to control trachoma, and this reduced mortality [122]. This, however, led to an increase in the efflux of resistance against AZI antibiotics [123]. There are various pieces of literature available against both Gram-positive and Gram-negative bacteria on the mechanism of efflux resistance against macrolides [124,125,126]. Abo-Zeid Y and her colleagues encapsulated AZI into PLGA polymers using the nanoprecipitation technique and studied the antibacterial activity of the AZI PLGA synthesized NPs against AZI resistance bacteria *Methicillin-resistant Staphylococcus aureus* (*MRSA*) and *Enterococcus faecalis* (*E. faecalis*), where one of the resistance mechanisms was resistance through the efflux pump [69]. They synthesized the AZI PLGA NPs with slight modifications from the method described by Mohammadi et al. [127]. To form the organic phase, 50 mg of PLGA and AZI (10, 15, 20 and 50 mg) were dissolved in 2 mL of acetone and the organic phase was added dropwise to 15 mL aqueous phase under sonication. The organic phase was evaporated by continuously stirring overnight at room temperature. This was followed by the collection of the NPs after centrifugation. The collected NPs were dried and stored in a desiccator. The synthesized AZI PLGA NPs exhibited a significant improvement in antibacterial activity when compared to free AZI. A four-fold decrease in the strain of MIC was observed in AZI PLGA NPs treated microbes compared to free AZI. The MIC values decreased from >1000 to 256 µg/mL and from 256 to 64 µg/mL for *E. faecalis* and MRSA, whereas the MIC value did not change for *P. aeruginosa*. It may be noted that *E. faecalis* and MRSA are known to possess resistance through the efflux pump, whereas for *P. aeruginosa* it was absent. They concluded that AZI PLGA loaded NPs were able to retain the antibacterial property of the AZI against efflux resistance bacterial species.

### 2.9. Magnesium

In 2003, silver and copper were used to study the antibacterial characteristics of metal nanoparticles; later that year, gold was the focus of this research. It should be mentioned that since then, further publications have been made about metal nanoparticles. There is available literature on the effectiveness of metal nanoparticles as antibacterial agents. In one of their research review articles, Gazzy and her colleagues reviewed the antibacterial potentials of metal-based nanocomposites in detail. They highlighted the mechanism of action, the antibacterial properties, and the various techniques used in the synthesis of these metal nanoparticles [128]. However, in this review, we will only talk about one element, magnesium, which is enclosed in PLGA-based nanoparticles.

Bone defects are usually at high risk of infection due to lack of soft tissue coverage, lack of blood supply, formation of local hematomas and also due to programmed cell death [129]. Surface material at the site of bone defect forms a hospitable environment for the growth or colonization of bacteria or microorganisms [130], resulting in the production of a polysaccharide—protein complex that favors the loading of the bacteria and results in biofilm formation [131]. As a well-known fact, biofilms are resistant to antibiotics and to the immune system of the host [132]. Infection at the site of the bone defects affects the blood supply and leads to osteonecrosis; this, in turn, severely hampers the recruitment of the osteoblast cells at the effected site by the immune cells. Delivery of the drug to the infected site is difficult, contributing to delayed healing or no healing at all. Using the low-temperature rapid prototyping (RP) technique, Ma et al. incorporated different Mg contents into the PLGA matrix to fabricate porous PLGA/Mg composite scaffolds and investigated antibacterial activity [133,134].

The low-temperature rapid prototyping method was used to create the porous PLGA/TCP scaffolds for the PLGA/TCP/Mg scaffold. In a nutshell, PLGA was dissolved into a homogenous solution in 1,4-Dioxane. Following that, TCP powders were incorporated into the PLGA solution at a 4:1 mass ratio. A cutting-edge low-temperature rapid prototyping machine was used to create the porous PLGA/TCP scaffolds at −30 °C. In accordance with the predesigned stereolithography model, all the porous scaffolds were spun layer-by-layer using a computer-driven nozzle to create precise 3D porous scaffold blocks that were afterward lyophilized. These scaffold were evaluated for their antibacterial activities [134].

Magnesium is an essential trace element required for the proper functioning of enzymes; Mg is a nonprotein enzyme helper. It is also essential for protein function. Magnesium is an essential trace element required by the human body. In the synthesis of proteins and nucleic acids, it plays a crucial role and is also involved in many cellular functions [135] due to its biocompatible, biodegradable and mechanical properties [136]. Mg and its alloys are used in orthopedic cardiovascular and ureteral stent applications [137,138]. Mg degradation contributes towards an increase in the pH that contributes towards its antibacterial properties [139,140]. As mentioned above, Ma et al. synthesized PLGA/Mg Scaffold using a low-temperature technique. They synthesized two different scaffolds using different content of Mg PLGA/10 wt.% Mg (PM-L) and PLGA/20 wt.% Mg (PM-H). They used PLGA scaffolds as a control. They used the same fabrication process as mentioned in their earlier studies [141]. The results indicated that the Mg-encapsulated scaffold caused an increase in pH because of the degradation and release of the Mg^2+^ ions. Antibacterial activity was exhibited by both PM-L and PM-H; they both inhibited the formation of biofilms. When PM-L and PM-H were compared, PM-H exhibited increased activity after 24 and 48 h of incubation compared to PM-L. From their result, it was evident that the PLGA/Mg scaffold exhibited significant antibacterial activity and that the higher concentration of Mg exhibited higher activity than the low content of Mg.

### 2.10. Natural Products and Anticancer Drugs

#### 2.10.1. Curcumin

Curcumin (*Curc*) [(E,E)-1,7-bis(4-hydroxy-3-methoxy-phenyl)-1,6-heptadiene-3,5-dione] is a bisα,β-unsaturated β-diketone. It is known for its potential biological activities such as antioxidant [142], anti-inflammatory [143], anti-diabetic [144], anti-carcinogenic [145], anti-angiogenesis and antimicrobial activities [146,147]. However, its therapeutical use has been limited due to its low bioavailability, pharmacokinetics, aqueous solubility and degradability at neutral to alkaline pH. Attempts have been made to enhance its biodegradability and solubility using biodegradable PLGA [148,149,150], but concerns regarding the toxicity of the NPs [148] have caused academics and researchers to look for an alternate using green synthesis. There are many sources available on green synthesis; A. Kumari et al. synthesized Curc-loaded PLGA NPs using plant extracts and assessed the biological activity of the synthesized NPs [151]. To synthesize Curc-loaded PLGA NPs, the leaf extract nanoprecipitation method was employed. They dissolved 100 mg of PLGA and 10 mg of Curc in 20 mL of acetone with continuous stirring for 2 h; to this solution, 20 mL of CSE was added and stirred for 10 h at 50 degrees Celsius. Acetone was removed using rotavapor; this was followed by centrifugation, and Curc@CSEPLGA NPs were collected. Similar conditions were used to synthesize Curc@DHEPLGA, Curc@REEPLGA and Curc@FPEPLGA NPs. To synthesize blank PLGA NPs plant extracts were used. Moreover, 100 mg of PLGA was dissolved in 10 mL of acetone. To this, 20 mL of CSE was added with continuous stirring at 50 °C for 10 h. The organic solvent was removed using rotavapor, and the CSEPLGA NPs were collected after centrifugation. To prepare blank DHEPLGA, REEPLGA and FPEPLGA NPs, the same method was used.

They concluded that among the plant extracts, *Camellia sinensis* was able to stabilize the NPs and enhance antibacterial activity. It also exhibited a slow and sustained release of curcumin. Curc@CSEPLGA NPs showed significant antibacterial activity against *E. coli* and *S. aureus*; these NPs also exhibited antibiofilm activity and were able to disrupt biofilm formation by bacteria in an effective way when compared to curcumin alone. They concluded that the synthesized NPs may be used as antibacterial and antibiofilm agents.

#### 2.10.2. *Alstonia scholaris*

Tripathi et al. synthesized and characterized the *Alstonia scholaris* leaf extract encapsulated in PLGA NPs by nanoencapsulation and evaluated the antibacterial activity of the synthesized ASPL NPs on both Gram-positive and Gram-negative bacteria [152] (Figure 5). They used green synthesis. The green approach to synthesizing NPs involves using plant base phytochemicals such as proteins, polyphenols, saponins, terpenoids, and flavonoids. These chemicals act as a reducing agent, stabilizer, and emulsifier for nanoparticles and enhance the efficiency of the nano-polymers. Plant-based NPs are preferred due to their availability, wide distribution, and the presence of various medicinal properties and lower side effects. *Alstonia scholaris* (L.) is commonly grown in the Indian subcontinent and in southeast Asia. *Alstonia scholaris* has grayish bark and milky sap. Studies have suggested that the plant is a bitter tonic with various medicinal properties and is used in the treatment of malaria, diarrhea, and dysentery. The leaf extract exhibited antimicrobial properties. Its use has an antimicrobial agent is limited due to poor bioavailability and water solubility [152]. The antibacterial efficiency of the leaf extract has been tested on Gram-positive bacteria such as *Bacillus subtilis*, *Staphylococcus albus*, *Staphylococcus aureus*, and Gram-negative bacteria such as *Escherichia coli*, *Klebsiella*, *Proteus vulgaris*, *Pseudomonas pyocyanea*, and *Shigella dysentriae*. However, as mentioned in the previous line, its use as an antimicrobial agent has been limited due to its poor bioavailability and poor water solubility (Table 2).

### 2.11. Antimicrobial Peptides (AMPs)

AMPs have emerged as one of the most interesting compounds with a broad spectrum of antimicrobial properties. Their antimicrobial properties are due to their broad spectrum of activity, multiple mechanisms of action, low resistance to bacteria, and significant clinical use [167,168]. AMPs are effector molecules from the innate immune system drawn from natural sources such as animals, plants, and microorganisms [168,169]. They are amphipathic with peptides with 30 amino acids and +2 and +9 charges due to the presence of high lysine, arginine and secondary structures [170,171,172]. The properties of these peptides interact with the cell membrane of pathogenic bacteria [173,174]. Due to their inactivation and disintegration in an acidic stomach environment, enzymatic action, and high ionic strength, their therapeutic application has been constrained. These limitations can be improved by using an alternate, such as loading the peptides and protein into NPs. Using NPs to encapsulate proteins and peptides can overcome the limitations. It may be noted that NPs exhibit low degradability, decreased toxicity, can be released to desired targets, are biocompatible with tissues and cells, and are suitable for oral administration with enhanced bioavailability [34,175,176,177].

Cruz et al. synthesized PLA and PLGA NPs loaded with the GIBIM-P5S9K peptide using the double emulsion solvent evaporation method and studied the in vitro antibacterial activity of the free peptide and the peptide-loaded NPs in *MRSA*, *E. coli* O157:H7 and *P. aeruginosa* [178]. Cruz et al. used solid-phase peptide synthesis to synthesize peptides [178,179]. They substituted the rink amide 4MBHA resin (100–200 mesh; loading: 0.63 mmol/g) and the FMOC amino acids [180] and applied the tea bag method, as described by Houghten R. A. [181]. These peptides were cleaved with the help of trifluoroacetic acid, (TFA)/tri-isopropyl silane (TIS)/ethanedithiol/H_2_O (92.5/2.5/2.5:2.5), for 2 h and then precipitated with cold diethyl ether [182]. For the synthesis of NPs, they used the method described by Cohen-Sela et al. with modification [183]. With the use of a homogenizer disperser, 1 mL of (1 mg/mL of GIBIM-P5S9K) peptide solution was dispersed in 4 mL of either dichloromethane (DCM) containing 10 mg PLA or 4 mL of ethyl acetate containing 10 mg PLGA to form the W1/O emulsion. Rotavapor was used to evaporate the solvents, such as ethyl acetate and DCM, from this first emulsion, which was mixed with 10 mL of 1% poloxamer 407 surfactant solution (W2 phase), then homogenized to form W1/O/W2 emulsion. Polyethyleneimine was added to ensure that the NPs had a positive charge. NPs were collected after centrifugation, washed, and freeze-dried. The peptide-loaded NPs showed higher antibacterial activity than the free peptides against *E. coli*, MRSA and *P. aeruginosa*. They concluded that these NPs loaded with peptides may be an alternative for the delivery of the GIBIM-P5S9K peptide, and these may also be protected from enzyme degradation.

## 3. Conclusions and Future Prospective

The biofilm structure of most microorganisms protects them against environmental hazards, including the host immune system and antimicrobial agents. This biofilm has recently led to the development of multidrug-resistant (MDR) strains. Conventional antimicrobial agents suffer from low penetration, high susceptibility to degradation, instability, and poor solubility in aqueous solutions. As a result, there is a huge demand for new antimicrobial drug classes or the development of new classes of drug nanocarriers to enable high selectivity and better localization. PLGA-based NPs are considered promising biocompatible, biodegradable drug delivery systems. In addition, PLGA-based NPs are easily synthesized with high drug loading capacity with different preparation protocols and can be decorated at their surface with active targeting ligands to selectively deliver their payload. Currently, researchers have developed several PLGA-based NPs loaded with antimicrobial agents to improve their effectiveness, minimize side effects, and enhance their prolonged release and active targeting via in vitro and/or in vivo evaluations. Thus, PLGA-based NPs could be considered as a potential candidate for the inhibition and destruction of microbial growth via improving the drug physiochemical. In the future, for the better utilization of this non-polymeric technology, scientists should emphasize understanding the mechanism of action of the PLGA-based NPs in vivo and consider the quality by design (QbD) manufacturing approaches to ensure a more rational design of these polymeric nanocarriers.

## Figures and Tables

**Figure 1 polymers-15-03597-f001:**
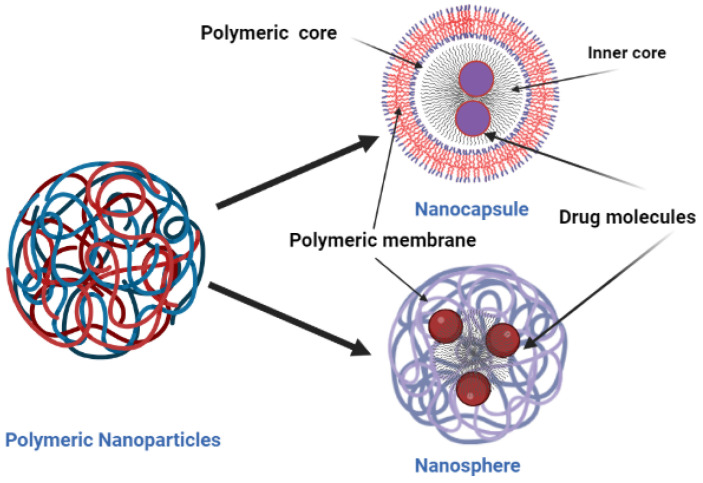
Schematic representation of the structure of nano-capsules and nanospheres.

**Figure 2 polymers-15-03597-f002:**
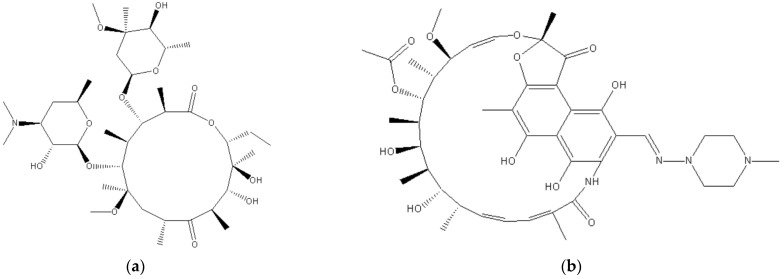
Chemical structures of (**a**) clarithromycin and (**b**) rifampicin.

**Figure 3 polymers-15-03597-f003:**
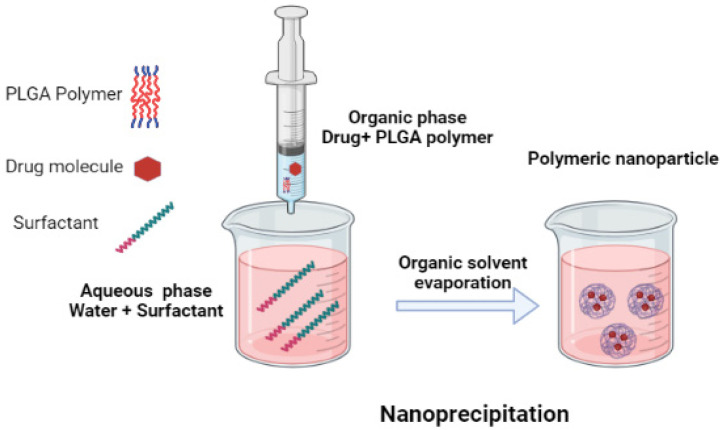
Schematic representation of the nanoprecipitation method.

**Figure 4 polymers-15-03597-f004:**
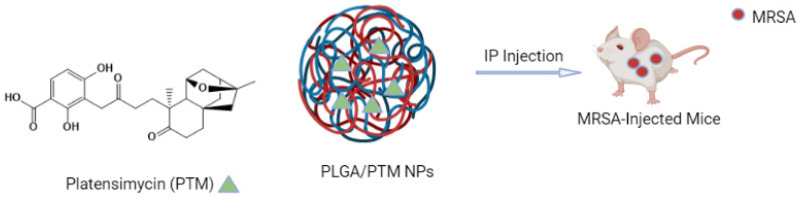
Overview of nano-strategy for the PLGA/PTM-loaded NPs.

**Figure 5 polymers-15-03597-f005:**
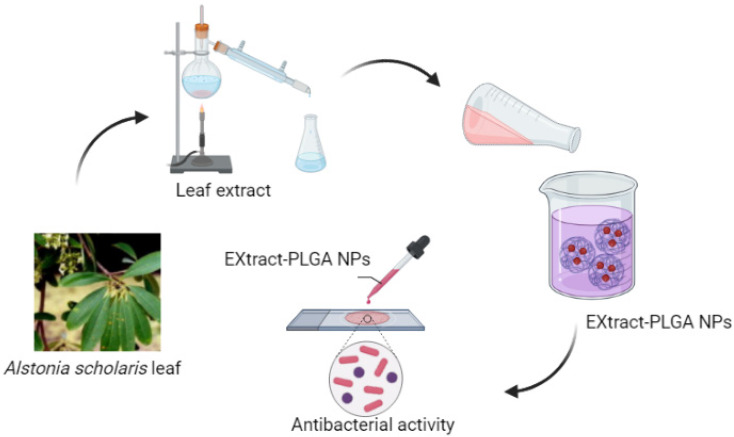
Schematic representation for *Alstonia scholaris* leaf extract encapsulated in polylactamide (PLGA) NPs.

**Table 2 polymers-15-03597-t002:** PLGA/PLA/Chitosan nanocomposites containing natural products and anticancer agents for biological application.

SN.	Polymers and Auxiliary Material Used			Antibiotics Used	Description of Formulation	Biological Activity	References	Year
1	PLGA	-	-	Plant extracts (as green surfactants), curcumin	Using green synthesis and nanoprecipitation, plant extract curcumin-loaded PLGA NPs are synthesized.	Improved solubility and photo stability	[151]	2020
2	PLGA	HPMC	-	Sweet potato leaves	Purple sweet potato leaf extracts loaded in PLGA submicroparticles to enhance the stability of flavonoids.	Antibacterial activity		
3	PLGA	-	-	Thyme oil	PLGA was coated with thyme oil by coacervation process.	Antimicrobial	[153]	2016
4	PLGA	Silk fibroin	-	Aloe anthraquinone	PLGA, silk fibroin, is an antibacterial agent from natural source, aloe anthraquinone (AA), blended by electrospinning for wound healing.	Wound healing	[154]	2022
5	PLGA	-	-	*Alstonia scholaris* leaf extract	Using solvent displacement method, *Alstonia scholaris* leaf extract loaded, PLGA NPs formulated.	Antimicrobial activity	[152]	
6	Cur-PLGA Complexes	Lactic acid (LA)	Glycolic acid (SSC and AAT conformers)	Curcumin	-	Antimicrobial activity	[155]	2021
7	PLGA	Dextran	-	Curcumin	-	Antibacterial activity	[156]	2021
8	PLGA	-	-	Curcumin (CUR)	To deliver the curcumin and increase its solubility and antibacterial properties, PLGA NPs were used.	Antibacterial activity	[157]	2020
9	PLGA	-	-	Curcumin	Encapsulation of curcumin in PLGA NPs.	Antioxidant	[158]	2015
10	PLGA	-	-	Epigallocatechin-3-O-gallate (EGCG)	By using electrospinning nanofibers method, EGCG, encapsulated PLGA sheets were prepared.	Prevention of post-surgical adhesion formation.	[159]	2015
11	PLGA	-	-	Ginseng/polyaniline	Microcapsules were equipped with antibacterial properties by incorporating Polyaniline (PANI) in ginseng-loaded PLGA microcapsules.	Antibacterial activity	[160]	2019
12	PLGA	Dextran	-	Cisplatin (CDDP)	PLGA dextran (PLD) non-delivery system was developed to improve the anticancer property of cisplatin.	Targeting of Cisplatin in Osteosarcoma	[161]	2015
13	PLGA-TH-NPs	-	-	Thymus vulgaris (thymol), amikacin antibiotics	Thymol-encapsulated PLGA NPs were studied for their synergistic activity in combination with antibiotics amikacin.	Antibacterial activity	[162]	2023
14	PLGA	-	-	-	Orthodontic adhesive (OQ) was incorporated into curcumin-loaded PLGA NPs.	Antimicrobial	[163]	2020
15	Poly(sarcosine) (PSar)	PEG	PLGA	Docetaxel	Docetaxel encapsulated poly(sarcosine) (PSar) and Polyethylene glycol (PEG) coated PLGA NPs.	Anticancer	[164]	2021
16	PLGA	-	-	Anticancer drug—doxorubicin (DOX) hydrochloride.	CNTs (carbon nanotubes) were used as nanocarriers and loaded with Doxorubicin (DOX) hydrochloride; then, these CNTs were encapsulated into PLGA nanofibers by electrospinning technique.	Antitumor Activity	[165]	2015
17	PLGA	Porphyrin photosensitizer sinoporphyrin sodium (DVDMS)	carboxymethyl chitosan sodium alginate	Basic fibroblast growth factor (bFGF)	Porphyrin photosensitizer sino porphyrin sodium (DVDMS) and PLGA encapsulated with bFGF nanosphere—embedded in carboxymethyl chitosan–sodium alginate to form hybrid hydrogel.	Wound healing.	[166]	2020

## Data Availability

Data available within the text.

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
