# Peer review of "Review on PLGA Polymer Based Nanoparticles with Antimicrobial Properties and Their Application in Various Medical Conditions or Infections"

_polymers, 2023, doi:10.3390/polym15173597_

Round 1
Reviewer 1 Report
The present review manuscript “Review on the Synthesis of PLGA Polymer Based Nanoparticles with Antimicrobial Properties and its Application in Various Medical Conditions or Infections” by Shakya et al. is novel and well-organized. The authors extensively reviewed different antimicrobial drug-loaded PLGA-based nanoparticles. Overall, the quality of the manuscript is good. However, there is much scope to improve the overall quality of the manuscript. My comments are as follows.
Comment 1. Title: In my opinion, the title of the manuscript is scientifically incorrect. There is no description of the synthesis/production of PLGA-based Nanoparticles. Therefore, the authors should delete the “synthesis of” from the title.
Comment 2. The authors should also discuss the PLGA, their properties, advantages, limitation, and modification by other polymers in a separate section.
Comment 3. What are the challenges in the delivery of antimicrobials? Kindly explain in a separate section.
Comment 4. Kindly provide an abbreviation list.
Comment 5. Kindly discuss some patents on antimicrobial drug-loaded PLGA-based nanoparticles.
Author Response
Please refer attachment, thank you

Reviewer 2 Report
Figure 1 is too similar to the Figure 4 in this paper: https://www.mdpi.com/1422-0067/22/12/6538
What is PNP on line 99?
Line 104 - 106 need references
I would specify whether it is diameter or radius when mentioning the size of NPs, e.g. 60 nm on line 193.
In part 2.5.2 Metronidazole, line 236-238, can you explain why it is not effective in vitro but effective under clinical studies?
On line 376, can you include the full name of RP technique?
On line 388, can you briefly introduce their fabrication process?
Some figures and tables are not mentioned in the text
Please try to improve the quality of English language, there are so many typo, run-on sentences, and grammar errors.
Author Response
Please refer attachment, thank you

Round 2
Reviewer 1 Report
The revision is satisfactory.
Reviewer 2 Report
Thank you for your response, I have no further comments.
The quality of English language was improved vs version 1